# A Polysaccharide from *Ficus carica* L. Exerts Immunomodulatory Activity in Both In Vitro and In Vivo Experimental Models

**DOI:** 10.3390/foods13020195

**Published:** 2024-01-07

**Authors:** Lin Ye, Qin-Qiu Zhang, Shang Lin, Qing Zhang, Jing Yan, Ding-Tao Wu, Shu-Xiang Liu, Wen Qin

**Affiliations:** 1College of Food Science, Sichuan Agricultural University, Ya’an 625014, China; yelinlyn1992@gmail.com (L.Y.); 2020318064@stu.sicau.edu.cn (Q.-Q.Z.); shanli@sicau.edu.cn (S.L.); zhangqing@sicau.edu.cn (Q.Z.); yanjing@sicau.edu.cn (J.Y.); sliu@sicau.edu.cn (S.-X.L.); 2Institute for Advanced Study, Chengdu University, Chengdu 610106, China; wudingtao@cdu.edu.cn

**Keywords:** structural characteristics, cytotoxicity, macrophage, functional activity, cyclophosphamide (CTX, Cy)

## Abstract

Polysaccharides from *Ficus carica* L. (FCP) exert multiple biological activities. As a biological macromolecule, the available knowledge about the specific structures and mechanisms of the biological activity of purified ‘Brunswick’ fig polysaccharides is currently limited. In the present study, chemical purification and characteristics were identified via chemical and instrumental analysis, and then the impact of FCP on immunomodulation activity in vitro and in vivo was examined. Structural characteristics showed that the molecular weight of the FCP sample was determined to be 127.5 kDa; the primary monosaccharides present in the FCP sample were galacturonic acid (GalA), arabinose (Ara), galactose (Gal), rhamnose (Rha), glucose (Glc), and xylose (Xyl) at a ratio of 0.321:0.287:0.269:0.091:0.013:0.011. Based on the investigation of in vitro immunomodulatory activity, FCP was found to stimulate the production of NO, TNF-α, and IL-6, and increased the pinocytic activity of macrophages. Further analysis revealed that FCP activated macrophages by interacting with Toll-like receptor 4 (TLR4). Moreover, the in vivo test results indicate that FCP showed a significant increase in serum pro-inflammatory factors in immunosuppressed mice. Overall, this study suggests that FCP has the potential to be utilized as a novel immunomodulator in the pharmaceutical and functional food industries.

## 1. Introduction

*Ficus carica* L., belonging to the Moraceae family, is a fruit-bearing tree that has been cultivated since ancient times and is native to Central Asia and the Mediterranean coastline of Europe. This Asian flowering plant species holds significant historical importance as one of the earliest cultivated fruit trees in these regions [1,2]. The commonly recognized name for *Ficus carica* L. is fig fruit, which has been widely consumed and utilized for both food and medicinal purposes for centuries [3,4]. Fresh fig fruit can be eaten both with and without peel. However, a large number of fresh figs are spoiled every year owing to physicochemical damage during the harvest and storage processes. In order to decrease spoilage during processing, fig fruit could be processed into a spirit beverage, fruit juice, dried fruit, or jam [5]. Figs are abundant in minerals, vitamins, amino acids, organic acids, phenols, flavonoids, and polysaccharides, which exhibit various biological activities including anticancer, antioxidant, anti-inflammatory, and immunological activities [5,6].

Natural polysaccharides are essential biological macromolecules that consist of more than ten monosaccharide molecules connected together by glycosidic bonds. These polysaccharides can be derived from diverse sources, including animals, plants, bacteria, and fungi [7,8,9,10]. Recently, natural polysaccharides have received significant attention due to their multitude of biological properties, including antioxidant, antiviral, hypoglycemic, anti-inflammatory, anti-tumor, and immunomodulatory effects, as well as their ability to regulate the intestine [11,12,13,14]. Likewise, polysaccharides from figs (FCPs) are among the bioactive compounds with beneficial potency and low toxicity. It has been reported that polysaccharides from figs have a good immunomodulatory activity on RAW 264.7 macrophages, and it has also been discovered that the pattern recognition receptor (PRR) Toll-like receptor 2 (TLR2) activates macrophages in response to FCPs via the nuclear factor-kappa B (NF-B) p65 signaling pathway [15]. The biological functions of polysaccharides are known to vary based on factors such as molecular weight, degree of branching, glycosidic bonds, and other chemical structures [16,17]. Consequently, the structural characteristics of polysaccharide forms need to be investigated to understand their biological activities. Nevertheless, available information on the precise structure and mechanisms underlying the biological activity of purified polysaccharides of *Ficus carica* L. is still limited. In particular, the mechanisms underlying their immunomodulatory activities remain largely unexplored. For example, whether Toll-like receptor 4 (TLR4) and its role as a pattern recognition receptor (PRR) participate in immunomodulatory activities or not still needs to be further investigated. Based on the existing knowledge gap, we have formulated a hypothesis that a novel polysaccharide derived from fig fruit may possess a distinct structure and exhibit significant immunomodulatory activity, and that Toll-like receptor 4 (TLR4) may be implicated in this biological processes.

Previous studies have reported that cyclophosphamide (CTX, Cy), a potent immunosuppressive agent, can result in immunosuppression in mice [18,19]. At present, there are few studies on the structural characteristics and immunomodulatory activity of FCP in vitro and in vivo. Therefore, the findings of our research offer valuable information about the intricate composition and biological properties of fig polysaccharides. Additionally, these results lay the foundation for food and functional products using polysaccharides extracted from *Ficus carica* L. Therefore, newly identified polysaccharides from figs are worthy of further research.

## 2. Materials and Methods

### 2.1. Materials and Reagents

‘Brunswick’ fig fruits were harvested at a commercial orchard (29°49′39.62″ N, 104°57′58.53″ E) belonging to Sichuan JINSIFANG Fruit Co., Ltd. Dried ‘Brunswick’ figs were bought from Sichuan JINSIFANG Fruit Co., Ltd. (Neijiang, China). Standard monosaccharides (galactose, Gal; fucose, Fuc; rhamnose, Rha; arabinose, Ara; glucose, Glu; fructose, Fru; xylose, Xyl; ribose, Rib; mannose, Man; galacturonic acid, GalA; glucuronic acid, GluA; galactosamine hydrochloride, GalH; glucosamine hydrochloride, GluH; N-Acetyl-D-Glucosamine, GluNAc; guluronic acid, GulA; mannuronic acid, ManA) and methylation reagent kit (BRT-JJH, AR) were acquired from Bo Rui Saccharide Biotech Co., Ltd. (Yangzhou, China). RPMI-1640 medium, penicillin, and streptomycin were purchased from Biosharp (Hefei, China). Fetal bovine serum (FBS) was acquired from ExCell Bio (Nanjing, China). Murine RAW 264.7 (ATCC TIB-71) macrophages were bought from American Type Culture Collection (ATCC; Manassas, VA, USA). Cell Counting Kit-8 (CCK-8) was acquired from SHANGHAI TAOSHU BIOLOGY SCIENCE AND TECHNOLOGY Co., Ltd. (Shanghai, China). Mouse TNF-α and IL-6 ELISA kit were acquired from Shanghai WellBio Technology Co., Ltd. (Shanghai, China). Griess reagent kit and neutral red were obtained from Beyotime (Shanghai, China). Bicinchoninic acid (BCA) protein assay kit was acquired from SolarBio (Beijing, China). Resatorvid (TAK-242) and lipopolysaccharide (LPS) were obtained from MedChemExpress (Monmouth Junction, NJ, USA). Cyclophosphamide (CTX, Cy) was bought from Jiangsu Hengrui Pharmaceuticals Co., Ltd. (Lianyungang, China). Remaining chemicals were all of analytical grade.

### 2.2. Polysaccharide Extraction and Purification

Polysaccharides were extracted according to our previously reported method [20] with minor modifications. Briefly, dried ‘Brunswick’ figs were ground with a pulverizer, passed through a 60-mesh sieve, and incubated in 100% (*v*/*v*) ethanol (1:10, *w*/*v*) at 60 °C for 1 h to remove most of the small molecule impurities three times. Then, the extract residues were used to extract water-soluble polysaccharides and deproteinization with Sevag reagent (CHCl_3_:CH_3_OH = 4:1, *v*/*v*) was performed according to the Sevag method [21]. The crude water-soluble polysaccharide solution was dialyzed against distilled water for 48 h and ultrapure water for 24 h (cut-off Mw 8000–14,000 Da). After ultrafiltration (cut-off Mw 100,000 Da), the filtrate solution was lyophilized as a pure polysaccharide, named FCP, for subsequent investigation.

### 2.3. Characterization of FCP

#### 2.3.1. Determination of Basic Components in FCP

The total carbohydrate, uronic acid, protein, starch, and total polyphenols were determined by the phenol–sulfuric acid colorimetric method [22], meta-hydroxydiphenyl method [23], bicinchoninic acid (BCA) protein assay, I-KI assay [24] and Folin–Ciocalteu method [25], respectively.

#### 2.3.2. Molecular Weight of FCP

The molecular weight (Mw) of FCP was measured by high-performance-size exclusion chromatography coupled with a multi-angle laser light scattering and refractive index detector (HPSEC-MALLS-RID) according to our previously reported method [26] with minor modifications. In brief, HPSEC-MALLS-RID measurements were performed using a multi-angle laser light scattering detector (MALLS, DAWN HELEOS, Wyatt Technology Co., Santa Barbara, CA, USA) coupled with an Agilent 1260 series LC system (Agilent Technologies, Palo Alto, CA, USA). Two columns, TSK-Gel G5000PWXL (300 mm × 7.8 mm, i.d.) and TSK-Gel G3000PWXL (300 mm × 7.8 mm, i.d.), were connected in series and maintained at a temperature of 30 °C. The mobile phase consisted of a 0.9% NaCl aqueous solution, and the sample was eluted using this mobile phase at a flow rate of 0.5 mL/min. The concentration of the sample was approximately 1.0 mg/mL, and an injection volume of 100 μL was used for the analysis [26].

#### 2.3.3. Monosaccharide Composition Analysis

A 5 mg sample of FCP was subjected to hydrolysis in a sealed ampoule using 2 mL of 3 mol/L trifluoroacetic acid (TFA) at a temperature of 121 °C for a duration of 3 h. The excess TFA was removed three times using nitrogen gas and methanol. Subsequently, the final hydrolyzate sample and standards were dissolved in ultrapure water and prepared for injection into the ion-exchange chromatography ICS-5000 system (Thermo Fisher, Sunnyvale, CA, USA). The system was equipped with a Dionex™ CarboPac™ PA20 column (150 mm × 3 mm, 10 μm) and an electrochemical detector [27].

#### 2.3.4. UV Analysis

The UV absorbance spectrum of the FCP (1 mg·mL^−1^) was analyzed based on an UV-1800PC spectrophotometer (MAPADA Instruments, Shanghai, China) with a spectrum range of 200 to 800 nm.

#### 2.3.5. FT-IR Spectroscopy Analysis

The Fourier-transform infrared spectroscopy (FT-IR) analysis of FCP (1 mg) mixed with potassium bromide powder at a range of 4000–400 cm^−1^ was performed on an FT-IR spectrophotometer Nicolet iS 10 FT-IR instrument (Thermo Fisher Scientific, Waltham, MA, USA) [28].

#### 2.3.6. SEM Analysis

The SEM of FCP was analyzed using a SU8100 scanning electron microscope (Hitachi, Tokyo, Japan). Briefly, the FCP powder sample was placed on a carbon-coated electrical film and coated with a sputtering of gold powder using a MC1000 sputter coater (Hitachi, Tokyo, Japan). Then, the FCP sample was imaged using magnifications of 500 and 5000 at an accelerating potential of 2 kV.

#### 2.3.7. Atomic Force Microscopy (AFM) Analysis

The morphology and molecular characteristics of FCP were analyzed using SPM-9700 atomic force microscopy (AFM) (SHIMADZU, Kyoto, Japan) in a tapping mode. Specifically, the FCP sample was completely dissolved in ultrapure water. Then, 10 μL of the FCP solution (5 μg⋅mL^−1^) was dropped onto a mica sheet and allowed to air dry at room temperature for 2 h. The prepared sample was then placed under the microscope for observation and analysis [29].

#### 2.3.8. Methylation Analysis

For the analysis of monosaccharide linkage, methylation was performed following the methods described by Ma et al. [30]. In brief, a 3 mg sample was completely dissolved in 1 mL anhydrous DMSO, followed by the addition of methylation reagents A and immediate sealing. After thorough dissolution through ultrasound treatment, methylation reagent B was added and the mixture was stirred for 60 min in a water bath at 30 °C. The methylation reaction was terminated by adding 2 mL ultrapure water. Subsequently, 1 mL of 2 M TFA was added to the methylated polysaccharide and the mixture was hydrolyzed for 90 min before being dried by rotary evaporation. To the dried product, 2 mL bi-distilled water and 60 mg NaBH_4_ were added and allowed to react for 8 h. The reaction was neutralized using acetic acid. After concentration by rotary evaporation and drying in an oven at 101 °C, 1 mL acetic anhydride was added for acetylation at 100 °C for 1 h. To remove acetic anhydride, 3 mL methylbenzene was added four times under rotary evaporation until dryness. The resulting product was then extracted with CH_2_Cl_2_ and dried with Na_2_SO_4_ for GC-MS analysis.

For GC-MS analysis, an Agilent 6890/5973 gas chromatography–mass spectrometer (Agilent Technologies, Santa Clara, CA, USA) equipped with an RXI-5 SIL MS column (30 m × 0.25 mm × 0.25 μm) was utilized. The temperature was programmed to increase from 120 °C to 250 °C at a rate of 3 °C⋅min^−1^ and held for 5 min. The injector and detector temperatures were set at 250 °C, and the helium flow rate was maintained at 1 mL⋅min^−1^.

#### 2.3.9. Triple-Helix Construction Analysis

The triple-helix structure of FCP was acquired based on the Congo red method in the previously reported literature [27]. In brief, the FCP sample with a concentration of 2 mg·mL^−1^ was mixed with the Congo red reagent at a concentration of 80 μmol⋅L^−1^ in a 1:1 (*v*/*v*) ratio. The NaOH concentration in the solution was incrementally increased from 0 to 0.5 mol⋅L^−1^. At each NaOH concentration, the maximum absorption wavelength was recorded during the process using a UV-1800PC spectrophotometer (MAPADA Instruments, Shanghai, China). A Congo red solution without FCP was used as a control for comparison.

### 2.4. Immunomodulatory Activity of FCP In Vitro

#### 2.4.1. Cell Culture, Grouping and Administration

RAW 264.7 macrophages were cultured in RPMI-1640 medium supplemented with 10% heat-inactivated FBS, 1% penicillin, and 1% streptomycin. All cells were incubated in a humidified 5% CO_2_ incubator at 37 °C [27].

The cells were seeded into a 96-well flat-bottom plate at a density of 3000 cells per well and incubated overnight. Then, FCP samples at different concentrations (FCP low-dose group, 100 μg⋅mL^−1^, FCP-L; FCP middle-dose group, 300 μg⋅mL^−1^, FCP-M; FCP high-dose group, 500 μg·mL^−1^, FCP-H;) were added to the wells. Apple polysaccharide (500 μg·mL^−1^, AP) and LPS (10 μg·mL^−1^) were used as the positive control. RPMI-1640 medium was used as the negative control (CK). To investigate whether Toll-like receptor 4 (TLR4) serves as the recognition site for FCP in macrophage activation, TLR4 inhibitor resatorvid (10 nM, TAK-242) and TLR4 inhibitor resatorvid (10 nM, TAK-242) with FCP (FCP high-dose group, 500 μg·mL^−1^, FCP-H) were explored. A blank (well without cells) was included. Three plates of each group were run simultaneously. The dosages of each group used above were determined based on previous experimental studies [15,29] with minor modifications.

#### 2.4.2. Cell Viability Assessment

RAW 264.7 cells were grown in 96-well plates at a density of 3000 cells per well and incubated overnight. Then, CK, FCP-L, FCP-M, FCP-H, AP, LPS, TAK-242 and TAK-242+FCP-H were added into the wells and the cells were incubated in a humidified 5% CO_2_ incubator at 37 °C for 48 h. Then, cells were incubated with 10 μL CCK-8 regent per well for 1 h at 37 °C. Cell viability was estimated using the CCK-8 assay [29] following the manufacturer’s instructions and the absorbance was measured using a DR-200Bc microplate reader (WUXI HIWELL-DIATEK INSTRUMENTS Co., Ltd., Wuxi, China) at 450 nm.

#### 2.4.3. Pinocytic Test

RAW 264.7 cells were grown in 96-well plates at a density of 3000 cells per well and incubated overnight. Then, CK, FCP-L, FCP-M, FCP-H, AP, LPS, TAK-242 and TAK-242+FCP-H were added into the wells and the cells were incubated in a humidified 5% CO2 incubator at 37 °C for 48 h. Then, 20 μL neutral red solution was added and thoroughly washed out with PBS after 2 h of incubation. Then, 200 μL of cell lysate solution was added and the plates were kept on a shaker for 10 min at room temperature. Finally, the absorbance was measured using a DR-200Bc microplate reader (WUXI HIWELL-DIATEK INSTRUMENTS Co., Ltd., Wuxi, China) at 540 nm.

#### 2.4.4. Determination of NO, IL-6, and TNF-a Cytokine Production Level

RAW 264.7 cells were grown in 96-well plates at a density of 3000 cells per well and incubated overnight. Then, CK, FCP-L, FCP-M, FCP-H, AP, LPS, TAK-242 and TAK-242+FCP-H were added into the wells and the cells were incubated in a humidified 5% CO2 incubator at 37 °C for 48 h. Then, the culture supernatant was collected from each well for NO, TNF-α and IL-6 analysis using a commercial Griess reaction kit, IL-6 kit, and TNF-α ELISA kit.

### 2.5. Immunomodulatory Activity of FCP In Vivo

#### 2.5.1. Animal Experiments

Twenty-eight male mice from the Institute of Cancer Research (ICR) were bought from Chengdu Dashuo Experimental Animal Co., Ltd. in Chengdu, China. These mice were specific-pathogen-free (SPF) and weighed approximately 30 g ± 2 g. They were housed in plastic cages with shavings under controlled conditions (a 12 h light/dark cycle; humidity: 70 ± 5%; temperature: 23 ± 2 °C) with ad libitum access to food.

After 1 week of adaptation, 10 mice were separated into a normal control group (CK). The remaining 30 mice were injected intraperitoneally with 80 mg/kg·bw/day of CTX for 3 days to induce immunosuppression [18]. The CK group was injected intraperitoneally with an equal volume of saline. These CTX-treated mice were then randomly divided into 3 groups (n = 10 per group): a model control group (MC), a low-dose group (FCP 50 mg/kg, LD), and a high-dose group (FCP 200 mg/kg, HD). In the LD and HD groups, mice were given 50 mg/kg and 200 mg/kg doses of FCP through intragastric gavage for 9 days, respectively. The CK and MC groups received a saline solution instead. The dosages of each group used above were determined based on previous experimental studies [18,19] with minor modifications. Animal care and use were approved by the Animal Care and Use Committee of Sichuan Agricultural University. The experimental design is shown in Figure 1.

#### 2.5.2. Determination of Serum Cytokines

After feeding for 9 days, all mice were anesthetized with diethyl ether and blood samples were subsequently collected. The serum was then collected and stored at 4 °C. The levels of interleukin (IL)-6 and tumor necrosis factor-α (TNF-α) in the serum were analyzed using ELISA kits in accordance with the instructions provided by the manufacturer.

### 2.6. Statistical Analysis

All experiments were conducted in triplicate unless otherwise stated. Data obtained were presented as mean ± standard deviation (SD). SPSS 26.0 software was used to test the normality of the data, and one-way analysis of variance (ANOVA) and Duncan’s test were used to analyze the differences between groups. *p*-value < 0.05 was considered to be statistically significant.

## 3. Results

### 3.1. Physicochemical Compositions and Molecular Weight of FCP

In this study, a crude fig polysaccharide was purified using the Sevag method and ultrafiltration. The content of the total carbohydrates was 93.46%, uronic acid was 60.40%, protein was 1.59%, and starch and total phenolics were not detected (Table 1). These results showed the fig polysaccharide had very low levels of impurities after being purified. In addition, the FCP sample had a molecular weight of 127.5 kDa and a polydispersity index (M_w_/M_n_) of 1.85, indicating the narrow molecular weight distribution of the purified FCP.

### 3.2. Monosaccharide Composition Analysis

The IC system analysis revealed the presence of the following constituent monosaccharides in FCP: galacturonic acid (GalA), arabinose (Ara), galactose (Gal), rhamnose (Rha), glucose (Glc), and xylose (Xyl). The molar ratios of these monosaccharides were found to be 0.321:0.287:0.269:0.091:0.013:0.011, respectively (Figure 2A).

### 3.3. Ultraviolet Spectrum Analysis

The protein content in the fig polysaccharide was found to be very low, which was consistent with the UV-Vis analysis indicating the absence of protein (<3%). This was confirmed by the absence of absorption peaks at 280 nm in the spectrum (Figure 2B) [31]. Furthermore, there was no absorption peak at 260 nm, which indicated that there was no nucleic acid in the fig polysaccharide (Figure 2B).

### 3.4. FT-IR Analysis

As shown in Figure 2C, two characteristic absorptions of the polysaccharide were observed at 3412 cm^−1^ and 2941 cm^−1^. These two strong absorption peak were O-H telescopic vibration and C-H asymmetric tensile vibration [32,33]. The absorption peaks of 1747 cm^−1^ and 1614 cm^−1^ were attributed to the C=O and carboxylate (-COO-) telescopic vibrations of carboxylic ester (-COOR), respectively, which also indicated the presence of uronic acid [34]. The absorption peaks of 1416 cm^−1^ and 1143 cm^−1^ were attributed to the bending vibrations of C-H or O-H and C-O-C, respectively, which indicated the presence of -OCH_3_ [35]; the C-O absorption of 1078 cm^−1^ and 1050 cm^−1^ indicated that the fig polysaccharide was of pyran configuration [29].

### 3.5. SEM Characterization of FCP

It is known that SEM can be used to detect the surface morphology of polysaccharides and that the images of SEM can indicate the molecular morphological features of polysaccharides [27]. The morphological characteristics of FCP were identified using SEM with magnifications of 500× and 5000× (Figure 3). At low magnification (500×) (Figure 3A), the polysaccharide was observed to have a smooth surface and lamellar structure, whereas at high magnification it had a distinct raised structure (5000×) (Figure 3B). These structures were relatively complete, indicating that the polysaccharides of each component were tightly polymerized and the interaction between molecules was strong, which may be related to the arrangement of polysaccharides of different molecules [32].

### 3.6. AFM Analysis of FCP

AFM is typically used to directly view macromolecular samples and provide two-dimensional images, but it is also used to directly observe three-dimensional surface images of polysaccharides with natural conditions [29]. Generally speaking, sugar chains with various compositions always have a propensity to form into the conformation that has the lowest free energy [31]. The fig polysaccharide showed a chain with a multiple-branch structure (Figure 4). These results suggest that FCP may first aggregate and then self-assemble into a long chain, with branching likely related to α-1,4-glycosidic linkages. This may be due to galacturonic acid, and is consistent with the methylation analysis [29].

### 3.7. Methylation Analysis

Methylation is a reliable method for determining the glycosidic bonds present in polysaccharides. In this study, the methylation reaction products of FCP were hydrolyzed and converted to their corresponding alditol acetates [36]. A total of 19 products were obtained and analyzed with GC-MS, as shown in Table 2. By comparing the obtained data with the standard data in the Complex Carbohydrate Research Centre (CCRC) spectral database for partially methylated alditol acetates (PMAAs) [37], the bonds of the monosaccharides in FCP were identified. These results agreed with the previously described analysis of the monosaccharide composition of FCP, showing a strong correlation between the two analyses.

### 3.8. Congo Red Analysis

In this study, the Congo red assay was used to investigate the presence of triple-helix structures in FCP [27]. It is known that polysaccharides forming triple helixes can cause a red shift in the maximum absorption wavelength (λ max) of the Congo red polysaccharide complex. The results showed that the maximum wavelength of UV-Vis absorption of the FCP + Congo red complex increased from 505 nm to 517 nm in the absence of NaOH (0.0 M) (Figure 5). This shift of λ max indicates the presence of a triple-helix structure in FCP [38].

### 3.9. Immunomodulatory Activity In Vitro

#### 3.9.1. Assessment of Cell Proliferation

The results shown in Figure 6A indicate that the tested concentrations of FCP did not have any impact on the toxicity of the cells. Moreover, the cells that were treated with FCP demonstrated a noteworthy enhancement in their viability when compared to the cells that were not treated. These results suggest that FCP does not induce cytotoxicity in RAW 264.7 macrophages and may even promote cell viability. At the same time, treatment with TLR4 inhibitor TAK-242 can markedly restrain cell viability, while FCP + TAK-242 can markedly promote cell viability.

#### 3.9.2. Assay of Phagocytosis

As shown in Figure 6B, FCP could enhance the phagocytosis of macrophages. This suggests that FCP has a stimulatory effect on the phagocytic ability of macrophages during immune response. At the same time, treatment with TLR4 inhibitor TAK-242 could markedly restrain phagocytic impact on macrophages, while FCP + TAK-242 could markedly stimulate phagocytic impact on macrophages.

#### 3.9.3. Determination of NO

The production of NO by macrophages is an essential defense mechanism that helps eliminate pathogens and control tumor growth. It underscores the critical role of macrophages in the immune system’s ability to fight infection and control tumor development [32]. After the treatment of the cells with FCP, the culture supernatants were collected and the nitrite content was determined (Figure 7A). From Figure 7A, it is evident that treatment with different concentrations of FCP significantly increases nitric oxide production in a dose-dependent manner. Moreover, treatment with TLR4 inhibitor TAK-242 could significantly decrease the release of nitric oxide in macrophages, whereas FCP + TAK-242 can significantly increase the release of nitric oxide in macrophages.

### 3.10. Immunomodulatory Activity In Vivo

As shown in Figure 8, the serum levels of IL-6 and TNF-α in the model group of mice were significantly reduced compared with those in the normal group; the high-dose group of fig polysaccharides significantly increased the serum levels of IL-6 and TNF-α in the CTX-induced immunosuppressed mice compared with the model group. These two results indicate that fig polysaccharides can improve the cellular immune function of CTX-induced immunosuppressed mice and have a strong immunomodulatory function.

## 4. Discussion

*Ficus carica* L. has recently attracted attention as a component of Chinese medicine and as a food for health purposes. It has anti-tumor effects, enhances immunity, and promotes digestion and appetite. Reports on the chemical composition and pharmacological mechanisms of ‘Brunswick’ *Ficus carica* L. are currently limited. This study aimed to address this knowledge gap by employing Sevag and ultrafiltration purification techniques to identify and characterize a polysaccharide from the plant which was subsequently designated as FCP. FCP is a heteropolysaccharide consisting of galacturonic acid (GalA), arabinose (Ara), galactose (Gal), rhamnose (Rha), glucose (Glu), and xylose (Xyl) with molar ratios of 0.321:0.287:0.269:0.091:0.013:0.011 and a molecular weight of 127.5 kDa. In vitro experiments revealed that FCP directly stimulated the activation of RAW 264.7 macrophages. In vivo data indicated FCP possessed an immune-enhancing effect in vivo to alleviate immunosuppression, which could be considered as a functional component and an immunological modulator in the food nutrition industry.

In a previous study, Du et al. reported the composition of polysaccharides from common *Ficus carica* L. This primarily consisted of five monosaccharides: rhamnose, arabinose, galactose, glucose, and mannose. The ratios of these monosaccharides were found to be 2.69:23.85:49.68:3.74:1.00, respectively [15]. Moreover, Chen et al. reported the presence of two polysaccharide fractions derived from *Ficus carica* L. The first fraction, FPs-1-1, primarily consisted of rhamnose, arabinose, xylose, mannose, glucose, and galactose, with molar percentages of 6.57:8.25:4.79:18.93:54.82:6.64, respectively [39]. The second fraction, FPs-2-1, primarily consisted of rhamnose, arabinose, xylose, mannose, glucose, and galactose, with molar percentages of 22.21:33.24:7.26:3.21:10.19:23.89, respectively [39]. The observed differences in the polysaccharide composition of *Ficus carica* L. in these results could be due to differences in extraction methods and the specific species used as a starting material. In our present study, the specific species of ‘Brunswick’ *Ficus carica* L. was used. However, the above two previous studies just purchased common figs from the local market without designating a specific species. The above two previous studies indicated that galactose had a relatively high content in fig polysaccharides, which is in line with our present results. We also found a triple-helical structure in FCP. These features could potentially be valuable for the chemical characterization and quality assurance of *Ficus carica* L.

Although the chemical properties of polysaccharides from different species of *Ficus carica* L. were different, most of them possess immune modulatory capacities. As is known to all, macrophages are monocyte-derived phagocytes that play a crucial part in the innate immune system’s defense and adaptive immune reactions because phagocytic cells function in the host immune system as regulatory and immune effector cells [40,41,42]. Polysaccharides from *Ficus carica* L. have been reported to increase proliferation and promote phagocytic function, the release of NO production, and the secretion of cytokines such as TNF-α and IL -6 by macrophages in vitro [15], which is in agreement with our results. After FCP treatment, proliferation, phagocytic function, NO production, and TNF-α and IL-6 level all increased dose-dependently. The Toll-like receptor (TLR) family plays a central role in mammalian immune response, constituting a significant part of the primary response mechanism to infections. Previous studies have reported that polysaccharides can activate macrophages through Toll-like receptor 2 (TLR2) and Toll-like receptor 4 (TLR4) [43]. In the present study, it was observed that FCP exhibited a symbolic impact on promoting proliferation, phagocytic function, NO production, and TNF-α and IL-6 level. Furthermore, it was found that FCP-induced macrophage activation was significantly inhibited in the presence of TLR4 inhibitors. This revealed that FCP activated RAW 264.7 cells by binding TLR4.

Cytokines are a class of small protein or glycoprotein molecules, secreted by activated immune cells or certain non-immune cells, that have a variety of biological functions [44]. IL-6 is a humoral immunomodulatory factor secreted by Th2 cells and is one of the most essential immunoinflammatory mediators that regulate various cellular functions such as B- and T-cell proliferation and differentiation [45].TNF-α is secreted predominantly by macrophages, although a number of other types of cells are also capable of producing it, inhibiting viral replication, and enhancing IL-2-mediated immunoglobulin production, NK cell activity, and monocyte proliferation [45]. The results of this experiment showed that, compared with the model control group, fig polysaccharide gavage intervention could alleviate the immunosuppressive state of mice by promoting the levels of IL-6 and TNF-α, thus exerting the function of regulating immunity. These results are consistent with previous findings [18,46].

As mentioned above, FCP showed a signification activating effect on macrophages. It could be applied as a new drug or nutrient food to enhance host immune systems in the future. However, to further investigate and develop FCP in terms of immunomodulatory activity, research should explore by which signaling pathway FCP activates macrophages and immunomodulatory activity, i.e., via MAPKs, NF-kB, or others signaling pathways. Although an animal experiment was carried out in this study, the issues of how fig polysaccharides are digested and absorbed in the body, and which genes are affected—thus affecting the body’s immune function—need further study. This study has laid a theoretical foundation for the quality control of polysaccharides. FCPs, as natural components, have many applications and can be used as excellent film-forming materials for food preservation. Quality control during food processing is an important research direction for the future.

## 5. Conclusions

In summary, the current study showed that FCP is a heteropolysaccharide from ‘Brunswick’ *Ficus carica* L. In the in vitro study, FCP showed a remarkable ability to increase phagocytosis and the proliferation of RAW 264.7 macrophages, and also to promote the secretion of nitric oxide (NO), TNF-α, and IL-6. In addition, a significant involvement of TLR4 in macrophage activation was detected. Moreover, in the in vivo study, CTX injection and classic modelling methods were assessed; the results indicate that FCP caused a significant increase in serum pro-inflammatory factors in immunosuppressed mice. These results suggest that FCP has the potential to address immune deficiency and exert immunomodulatory effects as a functional dietary supplement.

## Figures and Tables

**Figure 1 foods-13-00195-f001:**
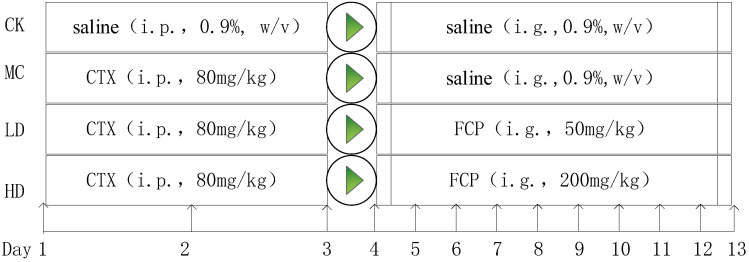
Schematic diagram for the experimental design.

**Figure 2 foods-13-00195-f002:**
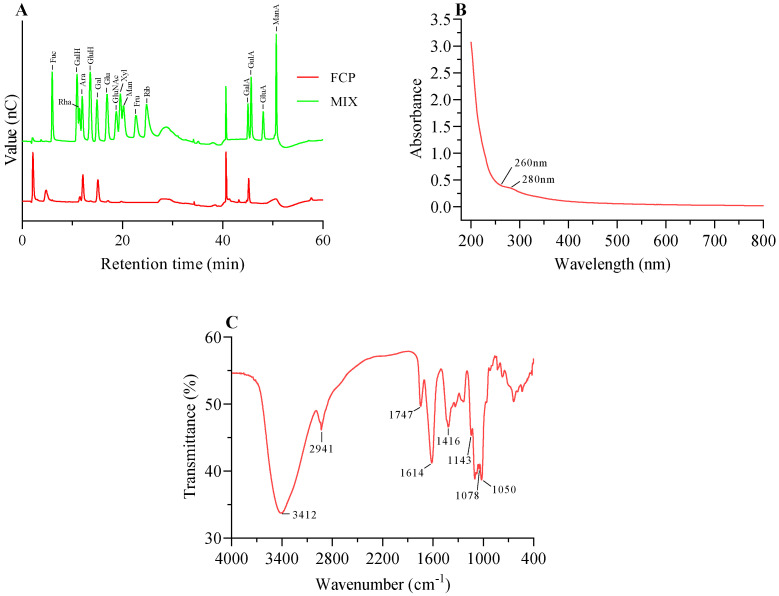
The monosaccharide composition, UV, and FT−IR analysis of FCP. (**A**) The monosaccharide composition of FCP. (**B**) UV analysis of FCP in the wavelength range of 200−800 nm. (**C**) FT−IR analysis of FCP.

**Figure 3 foods-13-00195-f003:**
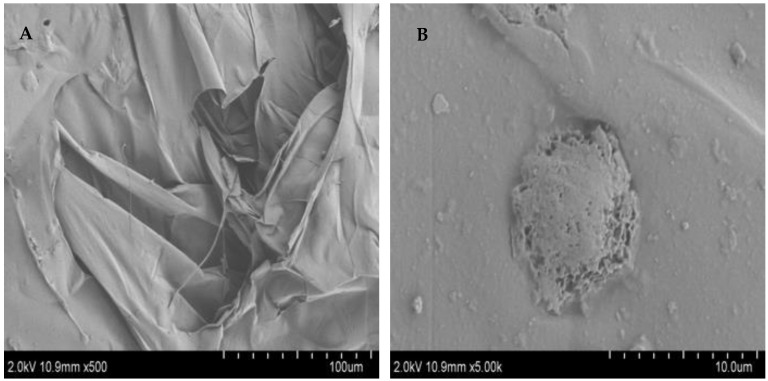
The SEM characterization of FCP. (**A**) Magnification at 500×. (**B**) Magnification at 5000×.

**Figure 4 foods-13-00195-f004:**
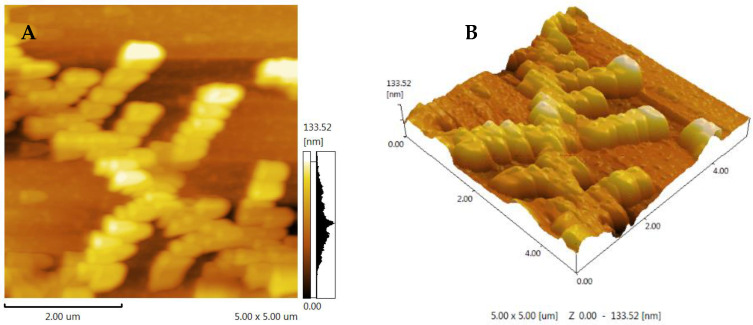
The AFM characterization of FCP. (**A**) Two-dimensional AFM images of FCP. (**B**) Three-dimensional AFM images of FCP.

**Figure 5 foods-13-00195-f005:**
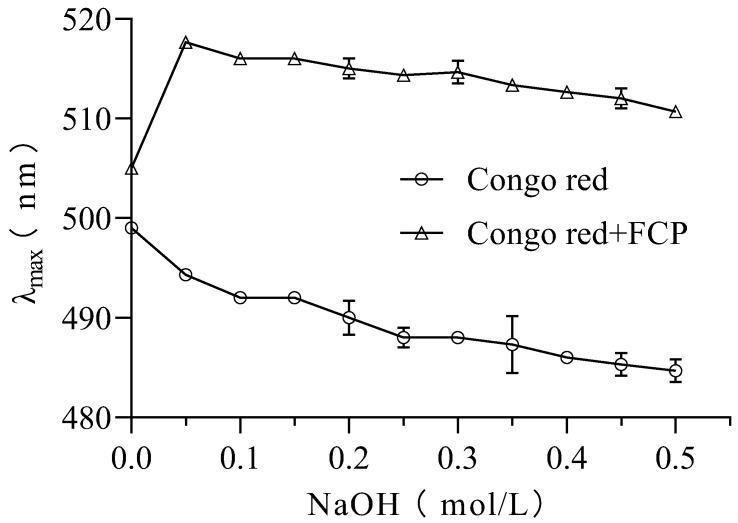
Congo red analysis of FCP.

**Figure 6 foods-13-00195-f006:**
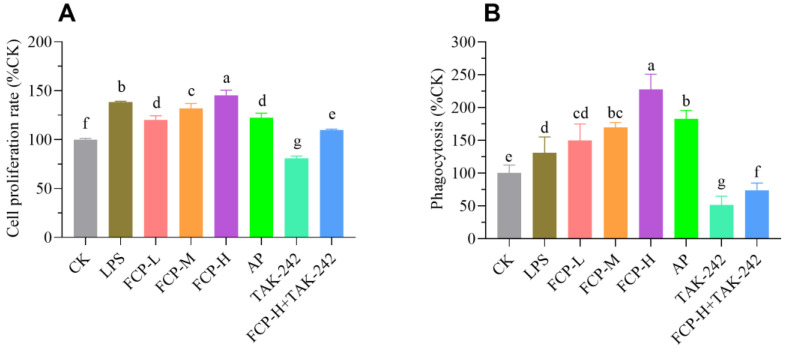
FCP activated macrophage RAW 264.7 cells. (**A**) The proliferation of the macrophage RAW264.7 cells treated with FCP. (**B**) The phagocytosis of macrophage RAW264.7 cells treated with FCP. One-way analysis of variance (ANOVA) and Duncan’s test were used to analyze the differences between groups. The different letters indicate statistically significant differences for each group *p* < 0.05.

**Figure 7 foods-13-00195-f007:**
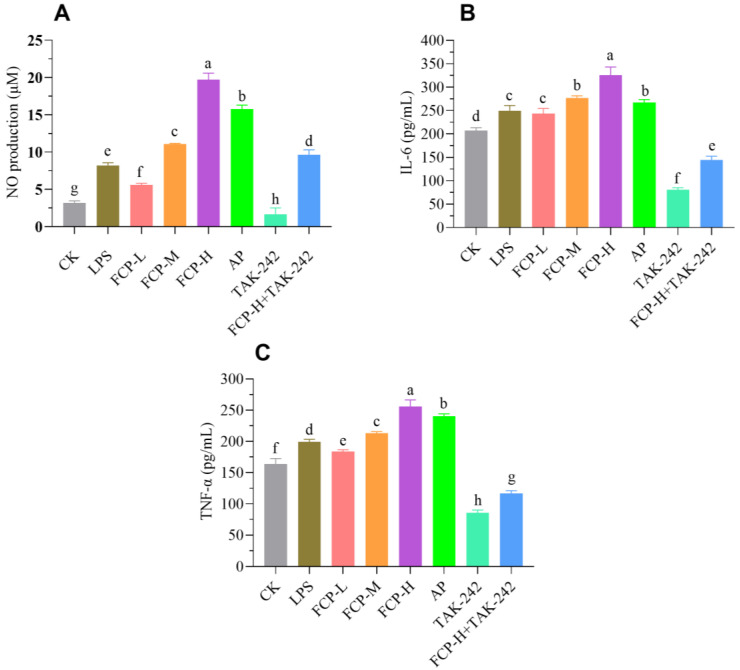
Effects of FCP on NO production and cytokines. (**A**) The NO production of the macrophage treated with FCP. (**B**) The secretion of IL-6 (**B**) and TNF-α (**C**) in macrophage treated with FCP. One-way analysis of variance (ANOVA) and Duncan’s test were used to analyze the differences between groups. The different letters indicate statistically significant differences for each group *p* < 0.05.

**Figure 8 foods-13-00195-f008:**
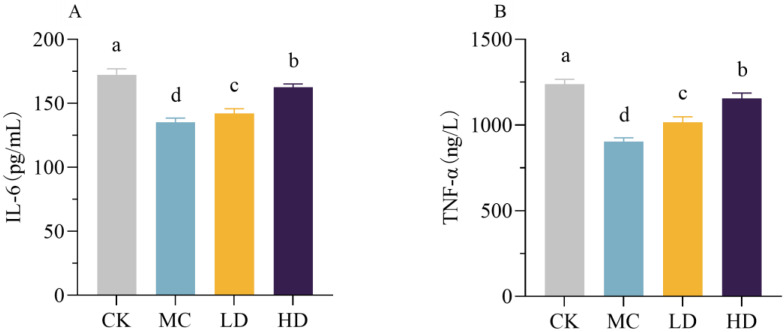
Effects of FCP on serum levels of proinflammatory cytokines in CTX-treated mice. (**A**) Interleukin-6, IL-6. (**B**) Tumor necrosis factor-alpha, TNF-α. CK, normal control group, injected intraperitoneally with saline and intragastric gavage of saline; MC, model control group, injected intraperitoneally with CTX and intragastric gavage of saline; LD, low-dose group, injected intraperitoneally with CTX and intragastric gavage of 50 mg/kg FCP; HD, high-dose group, injected intraperitoneally with CTX and intragastric gavage of 200 mg/kg FCP. One-way analysis of variance (ANOVA) and Duncan’s test were used to analyze the differences between groups. The different letters indicate statistically significant differences for each group *p* < 0.05.

**Table 1 foods-13-00195-t001:** The chemical composition and molecular weight of FCP.

Parameters	FCP
Total carbohydrate (%)	93.46 ± 2.27
Uronic acid (%)	60.40 ± 0.61
Protein (%)	1.59 ± 0.67
Starch (%)	n.d.
Total phenolics (%)	n.d.
Mw (kDa)	127.5
Mw/Mn	1.85

Mn: number-average molecular weight; Mw: weight-average molecular weight; n.d.: not detected.

**Table 2 foods-13-00195-t002:** GC-MS for methylated products of FCP.

RT	Methylated Sugar	Mass Fragments (*m*/*z*)	Type of Linkage
10.508	2, 3, 5-Me3-Araf	43, 71, 87, 101, 117, 129, 145, 161	Araf-(1→
11.804	2, 3, 4-Me2-Arap	43, 71, 87, 101, 117, 131, 161	Arap-(1→
13.997	2, 3, 4-Me3-Rhap	43, 59, 72, 89, 101, 115, 117, 131, 175	Rhap-(1→
15.223	2, 3-Me2-Araf	43, 71, 87, 99, 101, 117, 129, 161, 189	→5)-Araf-(1→
15.473	2, 4-Me2-Rhap	43, 58, 85, 89, 101, 117, 127, 131, 159, 201	→3)-Rhap-(1→
17.908	2, 3, 4, 6-Me4-Glcp	43, 71, 87, 101, 117, 129, 145, 161, 205	Glcp-(1→
18.541	2, 3, 4, 6-Me4-Galp	43, 71, 87, 101, 117, 129, 145, 161, 205	Galp-(1→
19.129	2-Me1-Araf	43, 58, 85, 99, 117, 127, 159, 201	→3, 5)-Araf-(1→
21.08	2, 4, 6-Me3-Glcp	43, 87, 99, 101, 117, 129, 161, 173, 233	→3)-Glcp-(1→
21.414	2, 3, 6-Me3-Galp	43, 87, 99, 101, 113, 117, 129, 131, 161, 173, 233	→4)-Galp-(1→
21.835	2, 4, 6-Me3-Galp	43, 87, 99, 101, 117, 129, 161, 173, 233	→3)-Galp-(1→
23.048	2, 3, 4-Me3-Glcp	43, 87, 99, 101, 117, 129, 161, 189, 233	→6-Glcp-(1→
23.757	2, 3, 4-Me3-Galp	43, 87, 99, 101, 117, 129, 161, 189, 233	→6)-Galp-(1→
24.424	2, 6-Me2-Glcp	43, 87, 97, 117, 159, 185	→3, 4)-Glcp-(1→
25.57	2, 6-Me2-Galp	43, 87, 99, 117, 129, 143, 159	→3, 4)-Galp-(1→
26.721	2, 3-Me2-Glcp	43, 71, 85, 87, 99, 101, 117, 127, 159, 161, 201	→4, 6)-Glcp-(1→
27.134	2, 4-Me2-Glcp	43, 87, 117, 129, 159, 189, 233	→3, 6)-Glcp-(1→
28.613	2, 3-Me2-Galp	43, 71, 85, 87, 99, 101, 117, 127, 159, 161, 201, 261	→4, 6)-Galp-(1→
30.602	2, 4-Me2-Galp	43, 87, 117, 129, 159, 189, 233	→3, 6)-Galp-(1→

## Data Availability

Data is contained within the article.

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
