# Peer review of "A Polysaccharide from *Ficus carica* L. Exerts Immunomodulatory Activity in Both In Vitro and In Vivo Experimental Models"

_foods, 2024, doi:10.3390/foods13020195_

Round 1
Reviewer 1 Report
Comments and Suggestions for Authors
These paper’s findings are encouraging and point out that polysaccharide from Ficus carica L, can be explored in future as promising immunomodulator in the pharmaceutical and functional food industry. Overall, I think the paper is interesting and need minor revision. The references are recent and the analyzes are robust and involve in vivo and in vitro experimental models. The suggestions are described below.
TITLE
I suggest a more authoritative title, which brings the main search result. Something like: Polysaccharides from Ficus carica L. exerts immunomodulatory activity in vitro and in vivo experimental models.
ABSTRACT
LINE 16 – “specific instruments” what kind of instrument? What kind of analysis? It is very generic.
LINE 17 – “in vivo and in vitro” in italic, please.
LINE 24 – “in vivo” in italic. Please check it throughout the text.
KEYWORDS
LINE 28 - Please replace the keywords by others different from the title to broaden the article search.
INTRODUCTION
LINE 41- Remove the word “etc” throughout the text. It is not formal.
LINE 42 – Exemplify some biological activities.
LINE 71-75 – Materials and Methods and not introduction.
LINE 77-80 Only about composition? What about biological properties?
MATERIALS AND METHODS
LINE 103 and 108 (for example): Please, add space between the reference and words throughout the text.
LINE 118 – Add “respectively”in the final of the sentence.
ITEM 2.38 and 2.3.9: Please, describe the methods briefly.
LINE 173 – 174: What is the meaning of FCP-L, FCP-M and FCP-H? Are they distinctive sample? L, M and H mean low, middle and high? Please, clarify it.
LINE 174 – Why was apple polysaccharide used as positive control?
ITEM 2.4.2, LINE 189 and 195– Please, remove “2.4.1” and describe the treatment.
ITEM 2.5.1 – Authors did not describe the group that received intraperitoneal injections of saline.
LINE 210: What doses?
LINE 211 – What is the meaning of NC?
RESULTS
LINE 313 – “On the cell viability.”
ITEM 3.9.1 - I do not agree when the authors pointed out tha FCP did not have any impact on cell viability. Depending on the type of treatment, there was an increase of almost 50% in viability. Were LPS, FCP-M, FCP-H really different from each other?
FIGURES: Please, place each bar on the graph with a different identification/color to differentiate the treatments.
FIGURE 8: Please, puto on the figure legend, the meaning of CK, MC, LD, HD, with your respective doses.
DISCUSSION
LINE 387: What kind of health purposes?
CONCLUSIONS
LINES 427-429: Unnecessary information for conclusion.
Reviewer 2 Report
Comments and Suggestions for Authors
Followings are my questions.
1. Please check uronic acid % in table 1.
2. Could you draw tentative structure of the polysaccharide?
3. Figure 6(A): TAK-242 showed cytotoxic effect. Is this OK?
Reviewer 3 Report
Comments and Suggestions for Authors
This paper (foods-2787588) described that FCP is a heteropolysaccharide from‘Brunswick’ Ficus carica L. with chemical properties in detail, as well as investigation of in vitro immunomodulatory activity of FCP to provide the fact disclosed to stimulate the production of NO, TNF-α, and IL-6 and increased the pinocytic activity of macrophages. Therefore, the reviewer recommended the revision as the following points out.
1) 1) Ficus carica is a well-known natural product, and its chemical composition varies depending on the region and climate. The reviewer understands that it was purchased from Sichuan JINSIFANG Fruit Co. But, detailed information on the place of origin should be provided.
2) In common with all figures showing biological activity, the statistical calculation method should be described in the legend of the figure.
3) As is common to all experiments showing biological activity, CK is treated as a positive control, but how the dosage is determined, including FCP, should be described.
Reviewer 4 Report
Comments and Suggestions for Authors
This research treated the new polysaccharides obtained from Ficus carica L. (FCP). The author determined the structural and chemical properties of FCP, as shown in Figs.2-4 and Tables 1 and 2. The addition of FCP induces the production of NO and regulatory effect of immune response such as IL-6, TNF-a, which implies that FCP modulates the immune system of cells. The present information is helpful for related researchers. Therefore, this manuscript need made some adequate revision, for examples, some points were claimed as listed below.
1) Line 13: Is the abbreviation of “polysaccharide from Ficus carica L.” FCP?
2) Line 50: Why can polysaccharides from figs be abbreviated by FCPs?
3) Lines 277-285: AFM images was a bit like the surface of salt-out effect of sodium chloride. How did the author confirmed that the image of AFM indicated the polysaccharides? The reviewer cold not understand from the explanation regarding the AFM measurement in 2.3.7 (lines 155-160).
4) Why the addition of FCPs altered the immune response of cell system? Although the present study had not focused on the point, the authors should give the insight or something else.
